# Piezoelectric Actuated Glass Plate for Liquid Density and Viscosity Measurement

**DOI:** 10.3390/mi11040348

**Published:** 2020-03-27

**Authors:** Baptiste Neff, Fabrice Casset, Arnaud Millet, Vincent Agache, Mikael Colin

**Affiliations:** 1CEA LETI, Univ. Grenoble Alpes, 38000 Grenoble, France; vincent.agache@cea.fr (V.A.); mikael.colin@cea.fr (M.C.); 2Team ATIP/Avenir Mechanobiology Immunity and Cancer, Institute for Advanced Biosciences INSERM U1209 CNRS UMR 5309, 38054 Grenoble, France; arnaud.millet@inserm.fr; 3Grenoble Alpes University, 38000 Grenoble, France; 4Department of Research, University Hospital of Grenoble Alpes, 38700 Grenoble, France

**Keywords:** Lamb waves, piezoelectric ceramics, viscosity measurement

## Abstract

This paper reports on a new system for liquid density and viscosity measurement based on a freely suspended rectangular vibrating plate actuated by piezoelectric ceramic (PZT) actuators. The Lamb mode used for these measurements allows us to infer both the density and viscosity in a larger range as compared to the existing gold-standard techniques of MEMS resonators. The combination of the measured resonance frequency and quality factor enables extraction of density and viscosity of the surrounding liquid. The system is calibrated while performing measurements in water glycerol solutions with a density range from 997 to 1264 kg/m^3^ and viscosity from 1.22 to 985 mPa·s, which is a larger dynamic range compared to existing mechanical resonators showing an upper limit of 700 mPa·s. The out-of-plane vibrating mode exhibits quality factor of 169, obtained in deionized water (1.22 mPa·s viscosity), and 93 for pure glycerol with a viscosity of 985 mPa·s. This Lamb wave resonating sensor can achieve measurement in fairly large viscosity media while keeping a quality factor superior to 90. Measurements performed on oil validate the use of the Lamb system. Oil density is evaluated at 939 kg/m^3^ and dynamic viscosity at 43 mPa·s which corresponds to our expected values. This shows the possibility of using the sensor outside of the calibration range.

## 1. Introduction

Micromachined vibrating systems are widely used as mass sensors for various applications. These systems could also be used in liquid, in order to measure physical properties like mass density and the viscosity of fluids. Measuring liquid properties is of great interest for the metrology of various kind of liquids like oil, ink, wine or milk [1,2,3,4]. It appears that mechanical resonating systems are well adapted to address this need as they obtain information on liquid properties thanks to vibrating parameters. Indeed, the damping arising from the vibration in a liquid medium depends directly on the liquid density and viscosity. It is well known that resonant frequencies and quality factors are the key parameters to characterize liquid damping. The damping in liquid results in two contributions: the first is the frequency shift due to the mass loading and the second is the quality factor decrease due to the viscous damping [5]. To investigate this field, mechanical resonators operating in a low kHz range have proven to feature a wider field of applications as compared to higher frequency resonators in the MHz range. The greater penetration depth in the millimeter range obtained in the low frequency regime can be leveraged to perform measurements in non-Newtonian fluids [6,7]. For liquids like polymer chains or surfactants, high frequency resonators might not be suitable because of their submicron penetration depth range [8,9]. The challenge is to decrease the resonance frequency while ensuring a relatively high quality factor, especially with high viscous loading.

Piezoelectric actuation authorizes to design complex resonators that will give the opportunity to generate specific modes of vibration. Cantilevers, micro-disks, circular membranes and rectangular plates are the most commonly used mechanical vibrating structures [10,11,12,13]. In order to optimize the measurement accuracy, we need mechanical resonators to exhibit the highest quality factor, higher than our reference criteria value of 100 in liquid evaluated from previous resonating sensors [14,15,16,17]. Previous studies have shown that in-plane and out-of-plane vibration microstructures can exhibit, respectively, a high quality factor of 300 in air or 100 in water. However, these values decrease drastically to 20 or 18 in commercial viscosity standard solutions with viscosity ranging up to 300 mPa·s [17,18]. Despite a better quality factor, in the case of in-plane vibrating structures, the frequency response is a function of the density and dynamic viscosity product. This dependence is problematic when the purpose is to determine these two parameters independently [19,20]. In comparison, flexural vibration modes seem to be more suitable for the determination of liquid properties, as vibrating parameters exhibit distinctive responses to density and dynamic viscosity density product [21]. Thus, for out-of-plane resonating sensor (like cantilevers), the viscosity of the surrounding liquid might be inferred from both determination of resonance frequency and quality factor. The main drawback of these modes is the important damping in liquid media due to a high-energy dissipation arising from inertial effect, acoustic radiation and viscous losses. Energy loss is even more important when resonant sensors are immersed in high viscous fluid, i.e., superior to 300 mPa·s, which affects the detection limit of this type of sensors. The resolution drops drastically as liquid viscosity increases, and as the measurement signal level decreases. To enhance the signal and increase the quality factor, the readout signal can be compensated with a static signal [22], while the quality factor can be improved by adding a Q-control unit [23]. To reduce the damping generated by the liquid, partial wetting might be also explored [24,25,26]. The idea is to limit the liquid-resonator interface in order to reduce the liquid damping. Recently, other out-of-plane modes have been studied and proved to be less impacted by viscous damping as opposed to cantilevers operating in flexural mode at moderate frequency [19,27].

It has been shown that Lamb waves are advantageous for stationary mechanical waves generated in liquid media and especially for sensing applications [28,29]. Antisymmetric Lamb waves exhibit a relatively high quality factor in liquid compared to other out-of-plane flexural modes of vibration due to a lower damping in liquid at low frequency [30]. At low frequency, the phase velocity of antisymmetric Lamb waves is subsonic which limits the acoustic radiation into the fluid [30,31]. The highest viscosity measurement obtained by this kind of mechanical resonator is at 700 mPa·s for a quality factor lower than 10 at a relatively low density of 871 kg/m^3^ [29].

In order to perform density and viscosity measurements of liquids in a larger range and in a cost-effective device, we devised a low-cost macroscopic sensor based on the generation of Lamb waves. Thus, we designed a freely suspended rectangular glass plate generating out-of-plane vibration with a high quality factor in liquid.

## 2. Materials and Methods

Our system is composed of a rectangular glass plate diced from a 200 mm glass wafer. Glass has the advantage of being a standard material used in common laboratories, with a lower cost as compared to silicon. Moreover, the Young modulus of glass is twice as low as the Young modulus for silicon, which allows us to decrease the resonance frequency for a given wavelength. The glass plate has a length of *L* = 40 mm and a width of *W* = 30 mm for a total thickness of *T* = 700 µm. To generate high order Lamb waves within the glass, 4 piezoelectric ceramics (APC International, Mackeyville, PA, USA) are used to actuate the plate. The actuators are made of 300 µm thick bulk PZT with silver electrodes (10 µm thick) patterned on both sides of the ceramics. The dimensions of the ceramics are 13 mm × 3 mm × 300 µm. Top electrodes of the actuators have been cut and separated in two distinct electrodes to facilitate the integration of the ceramics. The poling of the piezoelectric ceramics is perpendicular to the glass plate. Since the bottom electrode remains at a floating potential, the electric field applied to the 2 top electrodes creates an electric field in the same direction as the polarization of the piezoelectric material, resulting in a nodal line confined at the center of the ceramics. Actuators are arranged in two columns at specific locations corresponding to nodal lines of vibrations for the desired Lamb mode at each side of the plate to respect the symmetry of the Lamb wave and improve the propagation of the mechanical wave. The vibrating mode studied is the antisymmetric Lamb mode A0 with 12 nodal lines along the length of the glass plate (Figure 1c). The design rule to actuate this mode is to use actuators with a width smaller than half the wavelength λ/2 ≈ 3.5 mm in order to obtain optimal performances of the system.

Piezoelectric ceramics are placed using a plastic mold and then are glued to the plate using hard epoxy glue (UV15X-2, MasterBond, Hackensack, NJ, USA). The two actuators of each column are connected to each other with wedge bonding. In order to apply electric potential to the actuators flexible electrical connectors are used. To protect these electrical connections from the outside environment, the actuators are covered with silicone (MED 6010, NuSil, Carpinteria, CA, USA). The actuated glass plate is presented in Figure 1a. The plate is maintained using adhesive tape on top of small pillar at nodal points of the plate vibration mode. This method has been proven to not impact the wave pattern of the free plate [13]. Therefore, we can consider the rectangular structure to be freely suspended.

A housing fluidic cavity is micromachined out of a plexiglass plate (Figure 1b). The dimensions of the housing cavity are larger than the size of the glass plate to avoid any contact with the plate. The plate is placed upside down with the piezoelectric actuators facing the bottom of the cavity (Figure 1d).

## 3. Results

Density and viscosity measurements are performed for different water/glycerol ratios, from pure water to pure glycerol. To calibrate the prepared mixtures, viscosities are measured using a rheometer (Kinexus Pro, Malvern Panalytical, Orsay, France). All the measurements are performed at room temperature. The estimated density and dynamic viscosity of the tested liquids are presented in Table 1.

The response of the system in presence of liquid is measured using an impedance analyzer (IM3570, HIOKI, Nagano, Japan) with 500 mV applied to the electrodes. The actuators are both used for actuation and sensing by measuring the actuators impedance. To take advantage of the partial wetting of the resonator, 2 mL of liquid is dispensed on the top surface of the glass plate in order to cover completely the vibrating surface (Figure 1d). Actuation is performed on the four actuators to enhance mechanical Lamb wave propagation. After calibration of the impedance analyzer, impedance modulus and phase are measured for each liquid. The density of the tested liquids is ranging up from 997 kg/m^3^ to 1264 kg/m^3^ and the dynamic viscosity from 1.22 mPa·s to 985 mPa·s. Figure 2 shows the electric conductance spectrum for the Lamb mode with 12 nodal lines for different ratios of water/glycerol solutions.

From these measurements, and using the modified Butterworth–Van Dyke electromechanical equivalent model in liquid, we can extract the resonance frequency and the quality factor.

In the lumped parameter model from Figure 3a [14], *R*_0_ represents the dielectric loss of the actuators, *C*_0_ is the piezoelectric capacitance, *R* the connection resistance, *C_m_*, *L_m_* and *R_m_* are related to the electromechanical transduction in vacuum. The other parameters *R_v_* and *L_v_* are related to the viscous fluid loading on the plate which correspond respectively to the added damping and the added mass.

From Nyquist representation of the measured impedance in Figure 3b, we can determine the quadrantal frequencies of the resonance peak and thus extract the quality factor of the resonance from Equation (1) [32].
(1)$$Q=\sqrt{\frac{f_{q1}f_{q2}}{{\left(f_{q1}-f_{q2}\right)}^{2}}}$$
where $f_{q1}$ and $f_{q2}$ are the two quadrantal frequencies. Extracted resonance frequencies and quality factors are presented in Table 2. The highest quality factor of 163 is obtained in deionized water and it slightly decreases to 93 for a dynamic viscosity of 985 mPa·s. These obtained quality factors are unprecedented at such viscosity levels even compared to torsional PZT tube in pure glycerol which exhibits a quality factor of 63 [14]. Thus, we can expect to increase the calibration range to measure higher viscosity liquids.

## 4. Discussion

To evaluate the liquid properties of a fluid of unknown properties, we calibrate our vibrating system assuming the model of an oscillating sphere which has been proven to work with different resonator geometries and vibrations operated in liquid [33,34,35]. In this generalized model, the resonance frequency and the quality factor are related to the additional damping and added mass due to the presence of liquid. The additional damping and the added mass are linked to $g_{1}$ and $g_{2}$ expressions, respectively. These parameters can be recapitulated in the expressions describing the mechanical behavior of the vibrating plate, as followed:(2)$$f_{n}=\frac{1}{2\pi \sqrt{g_{2}}}$$
(3)$$Q=\frac{\sqrt{g_{2}}}{g_{1}}$$
where, $f_{n}$ is the resonance frequency of the vibrating plate and *Q* its quality factor for the corresponding Lamb mode. $g_{1}$ and $g_{2}$ are frequency-dependent functions depending on the liquid density and viscosity and on six arbitrary coefficients $C_{0}$, $C_{1}$, $C_{2}$, $C_{3}$, $C_{4}$ and $C_{5}$.
(4)$$g_{1}=C_{0}+C_{1}\sqrt{f_{n}}\sqrt{\rho \eta }+C_{2}\eta$$
(5)$$g_{2}=C_{3}\rho +C_{4}\sqrt{\rho \eta }/\sqrt{f_{n}}+C_{5}$$

These coefficients are obtained by measurements on calibrated liquids and after least squares optimization performed on MATLAB. The optimal values are found: $C_{0}=9.19\times 10^{-9}\;s$, $C_{1}=2.52\times 10^{-14}\;\frac{m^{2}s^{2}}{kg}$, $C_{2}=-1.39\times 10^{-12}\;\frac{m\;s^{2}}{kg}$, $C_{3}=5.75\times 10^{-16}\;\frac{m^{3}s^{2}}{kg}$, $C_{4}=9.17\times 10^{-15}\;\frac{m^{2}s^{2}}{kg}$, $C_{5}=1.79\times 10^{-12}\;s^{2}$.

From the values obtained for the calibration coefficients, it appears that in our range of density and viscosity tested, the second term of $g_{2}$ expression (related to the constant $C_{4}$) is negligible. The same conclusion can be drawn in the third term in $g_{1}$ expression, as it is also negligible. The resonance frequency and quality factors are plotted in Figure 4a,b and are compared with the analytical model presented in Equations (2) and (3). It shows a good fitting between experimental values and analytical model. As shown in Figure 4a, resonance frequency decreases linearly with liquid density. For density range from 997 kg/m^3^ to 1264 kg/m^3^, resonance frequency decreases from 103,376.7 Hz to 99,763 Hz. Values of 1/*Q* appear to be almost linearly proportional with the square root of the density viscosity product (Figure 4b). These dependences confirm that the generalized analytical model of the oscillating sphere comprising six calibration terms can be reduced to four terms in the case of an out-of-plane vibrating plate. Error bars for experimental values shown in Figure 4a,b are determined by repeating the measurement five times on the same tested liquid after the cleaning of the plate.

The model given in Equations (2) and (3) can be inverted to infer the density and viscosity as a function of the resonance frequency and the quality factor. The Lamb resonator results are compared with values obtained by rheometry and binary mixture model in Figure 5 [33]. The calibrated system is tested on oil with a known measured viscosity of 52 mPa·s and expected density of 925 ± 15 kg/m^3^. The extracted density is 939 ± 9 kg/m^3^ and dynamic viscosity is 43 ± 12 mPa·s which confirm the use of Lamb resonator beyond calibration range to be able to measure all kind of liquid with density lower than 997 kg/m^3^. Figure 5 also shows the data collected from oil outside of the calibration curve. Measurements at higher viscosity seem also to be achievable thanks to the high quality factor of 93 obtained at the upper viscosity limit of our calibration.

## 5. Conclusions

In conclusion, we presented a sensor based on a vibrating glass plate in order to enable the measurement of viscosity and density of liquids with high precision, and over a large dynamic range (997 kg/m^3^ to 1264 kg/m^3^ in density and 1.22 mPa·s to 985 mPa·s in viscosity). The parameters are inferred from the frequency response of the plate actuated by a piezoelectrical transduction. The extracted resonant frequencies and quality factor results were compared with estimated results from an analytical model of an oscillating sphere in liquid. It shows that resonant frequency is mainly dependent on the density of the liquid, whereas the quality factor depends on the density–viscosity product.

From results obtained in water to pure glycerol, resonance frequency decreases from 103,376.7 Hz down to 99,763 Hz and the quality factor decreases from 163 to 93 on a large scale of density. For high viscosity, the quality factor remains at relatively high level. After calibration, the first measurement is completed on oil, which validates Lamb resonant sensors for viscosity and density evaluation on a larger range than our calibration. From the trend presented by Figure 4b and the calibration of the oscillating sphere model, for a viscosity standard S600 (ρ=867 kg/m^3^; η=2063 mPa·s at 20 °C) we expect to obtain a quality factor of about 87 in this more viscous fluid which is promising for further measurements at higher viscosity.

## Figures and Tables

**Figure 1 micromachines-11-00348-f001:**
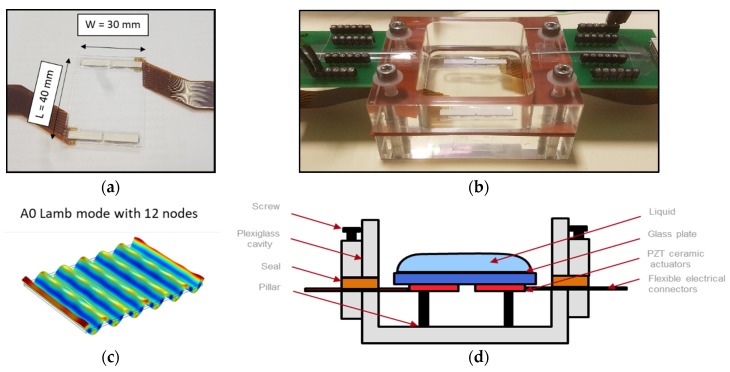
(**a**) Front view of the rectangular glass plate with the four ceramic actuators, (**b**) Photography of the vibrating plate inside the fluidic packaging with electrical connections, (**c**) 3D image obtained by finite element method (FEM) of the antisymmetric Lamb mode with 12 nodes, (**d**) Schematic cross section of the glass actuated plate inside a fluidic cavity.

**Figure 2 micromachines-11-00348-f002:**
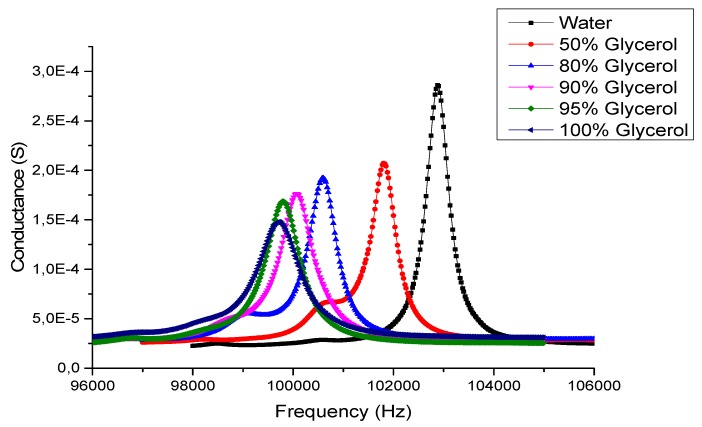
Conductance of the four piezoelectric actuators for different water/glycerol ratios obtained at 500 mV.

**Figure 3 micromachines-11-00348-f003:**
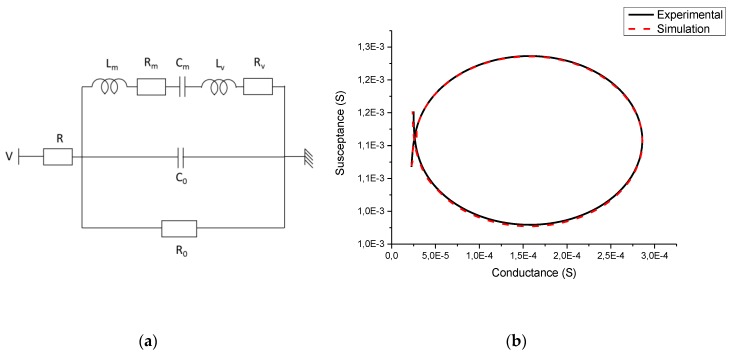
(**a**) Modified Butterworth–Van Dyke equivalent circuit in liquid [14]; (**b**) Nyquist representation of the impedance around the mechanical resonance in deionized water, experimental measurement and lumped model representation.

**Figure 4 micromachines-11-00348-f004:**
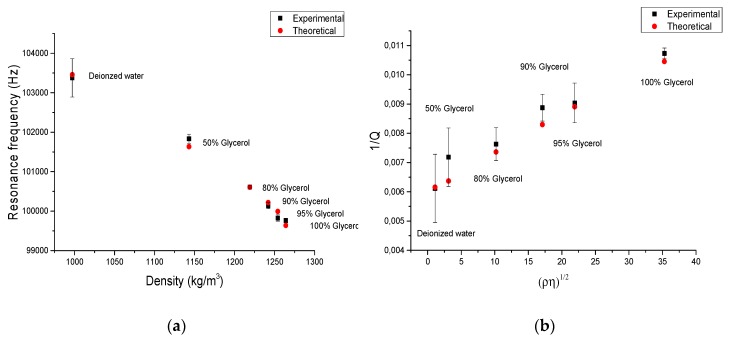
(**a**) Evolution of the resonance frequency obtained with the tested liquids in function of liquid density, (**b**) evolution of the quality factor in function of the density viscosity product, experimental and theoretical.

**Figure 5 micromachines-11-00348-f005:**
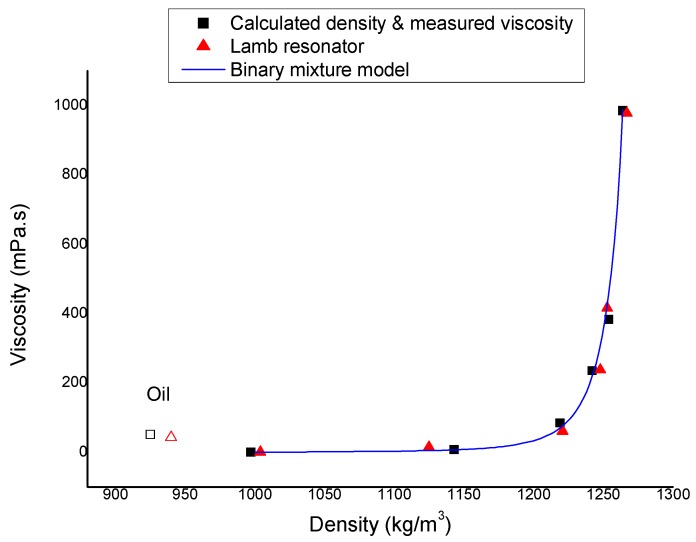
Comparison between density and dynamic viscosity measured by the Lamb resonator, rheometer results and binary mixture model for different water–glycerol mixtures and data extracted from oil outside of the calibration curve.

**Table 1 micromachines-11-00348-t001:** Density and measured dynamic viscosity of tested liquids (Rheometry).

Tested Liquids	Calculated Density (kg/m^3^)	Measured Dynamic Viscosity (mPa·s)
Water	997.1	1.22
50% Glycerol	1143	8.5
80% Glycerol	1219	85.3
90% Glycerol	1242	236
95% Glycerol	1254	383
100% Glycerol	1264	985

**Table 2 micromachines-11-00348-t002:** Extracted resonance frequencies and quality factors for each tested liquid.

Tested Liquids	Resonance Frequency (Hz)	Quality Factor
Water	103,376 ± 485	163 ± 31
50% Glycerol	101,833 ± 25	139 ± 19
80% Glycerol	100,610 ± 75	131 ± 10
90% Glycerol	100,140 ± 79	113 ± 6
95% Glycerol	99,825 ± 43	111 ± 8
100% Glycerol	99,763 ± 86	93 ± 2

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
