# Peer review of "Piezoelectric Actuated Glass Plate for Liquid Density and Viscosity Measurement"

_micromachines, 2020, doi:10.3390/mi11040348_

Round 1
Reviewer 1 Report
In their manuscript Neff et al. demonstrate a sensor for density and viscosity measurements in highly viscous fluids. The sensor comprises a relatively large glass plate with piezoelectric actuators which excite antisymmetric Lamb waves in the 100 kHz regime in the plate. The authors expose the sensor to several mixtures of water and Glycerol and electrically characterize the resonator. From the inferred resonance frequencies and quality factors the authors fit the parameters of a sphere model to determine density and viscosity of the fluid the sensor is exposed to. After model calibration the density and viscosity of a known oil are measured to verify the model.
The manuscript is well written and technically sound. I generally enjoyed reading it and appreciate the authors’ effort to propose and demonstrate a fluid sensor for highly viscous fluids. However, there are some issues and open questions such that the manuscript does not meet the criteria for a scientific publication yet.
- In the introduction the authors state the frequency should be as low as possible. Is this true? If yes why? Or did the authors rather mean that the penetration depth should be significant?
- The authors should detail on the statement that the quality should at least be 100 to determine fluid properties? Is this a fundamental limit? What are the underlying assumptions? A reference does not seem to be sufficient here.
- In line 55, the authors should note that also for flexural modes density and viscosity cannot be determined independently?
- The authors propose a very specific design. Why did they choose this design and no other design? Why do they use glass and not for example silicon?
- The authors claim that the resonator is freely suspended based on theoretical considerations? Can they check is this is really the case in experiment?
- Why is the suspension required? In highly viscous fluids anchor should not be dominant.
- Why is the sensor not fully immersed in a liquid?
- In figure 1, scalebars would be helpful.
- What is the oscillation amplitude the is reached?
- Is the targeted Lamb mode chosen because it is convenient to excite or is there another reason?
- My main points of critique regard the theoretical model. One of these points is that it seems to be fairly brave to approximate a thin rectangular plate with a sphere. Is the any justification for this approximation?
- On the other hand, the model is based on six(!) parameter. In that sense it is not surprising that from fitting it to six different fluids the resulting curve matches the experimental data quite well (see also https://doi.org/10.1119/1.3254017). Are all parameters equally important? The number of parameters should be reduced for a meaningful model to obtain predictive power.
- In the same manner, the verification fluid is not chosen appropriately. In figure 5, the corresponding point would be very close to the water point. So it is not surprising that there is good agreement between the measured and the reference values. Why did the authors not choose a point that is further away from the curve used for calibration? This would be much more convincing for testing the predictive power of the method.
- The authors claim that they have a sensor which is suitable for highly viscous fluids but the fluid used for model verification has a viscosity close to water. How well does the method work for highly viscous fluids?
- In figure 4, the authors could consider a 3D scatter plot to show the dependence on both density and viscosity.
I am confident that the authors can resolve these issues and I would like to encourage them to continue with their interesting and relevant work.
Author Response
Dear Editor, and Reviewers
Please find attached the revised version of our manuscript entitled, “Piezoelectric actuated glass plate for liquid density and high viscosity measurement,” which we submitted for publication in Micromachines.
We thank the Reviewers for their insightful comments, which we believe have significantly improved our manuscript. We have revised the manuscript in accordance with these comments and responded to all queries in a separate document entitled, ‘Response to Referees1’. All changes to the text are shown in blue.
Please the attachment.
Thank you for your consideration.
Sincerely yours,

Reviewer 2 Report
See attached file

Author Response
Dear Editor, and Reviewers
Please find attached the revised version of our manuscript entitled, “Piezoelectric actuated glass plate for liquid density and high viscosity measurement,” which we submitted for publication in Micromachines.
We thank the Reviewers for their insightful comments, which we believe have significantly improved our manuscript. We have revised the manuscript in accordance with these comments and responded to all queries in a separate document entitled, ‘Response to Referees2’. All changes to the text are shown in blue.
Please see the attachment.
Thank you for your consideration.
Sincerely yours,

Round 2
Reviewer 1 Report
I thank you the authors for their reply to the referee questions. Most of the questions have been answered in a satisfactory manner. However, the question about the experimental verification and the main message of the paper remains open. There are two reasons for this.
First, it is clear that water is intuitively different from oil which has been used for testing the model. However, the authors should put the corresponding data point in figure 5. In figure 5, the data point would not be far away from the curve defined by the tested fluids. The authors should explain their choice of the test fluid (are there alternatives?) and discuss if the obtained results are also achievable for fluids further away from the calibration curve.
Second, if the main purpose of the paper is to present a sensor for highly viscous fluids (as claimed in the title), why is the test fluid a low viscosity fluid? Can a quick experiment be performed with for example honey as the authors propose? I agree that a detailed study of highly viscous fluids is beyond the scope of the manuscript but a simple experimental demonstration would certainly strengthen the claimed results.
I am sure that the authors can either include these data easily or reformulate the corresponding claims in the manuscript. After that the manuscript should be ready for publication.
Author Response
Thank you for your review, please see the attachment.

Reviewer 2 Report
Dear Authors,
Thank you for the extensive answering of my questions. I regret that you did not use my theoretical description of your system, but maybe you want to confine to the experimental results and the calibration of the system by standard fluids. I have no problem with that. Good luck!
Author Response
Thank you for your review.